# Evaluation of the Abbott Panbio™ COVID-19 Ag Rapid Antigen Test for Asymptomatic Patients during the Omicron Wave

Viet Tran [1,2,3,*] , Giles Barrington [1,2] , Zach Aandahl [4] , Amelia Lawrence [2], Senudi Wijewardena [2], Brian Doyle [1,2] and Louise Cooley [1,2]

1 Royal Hobart Hospital, Tasmanian Health Service, Hobart 7000, Australia
2 Tasmanian School of Medicine, University of Tasmania, Hobart 7000, Australia
3 Menzies Institute for Medical Research, University of Tasmania, Hobart 7000, Australia
4 School of Natural Sciences, University of Tasmania, Hobart 7000, Australia
* Correspondence: v.tran@utas.edu.au

**Abstract:** Rapid antigen testing (RAT) is a cost-effective and time-efficient method of identifying severe acute respiratory syndrome coronavirus 2 (SARS-CoV-2) and therefore a critical part of infection control strategies. There is no published evidence evaluating the use of RAT during the Omicron wave of the COVID-19 pandemic for asymptomatic patients or its performance between waves. All patients presenting to an Emergency Department over a two-week period without COVID-19 symptoms were screened for SARS-CoV-2 using both the Abbott Panbio RAT as well as the gold standard reverse transcriptase real-time polymerase chain reaction (PCR). The Abbott Panbio RAT sensitivity was 13% (95% CI 0.028, 0.336) for asymptomatic patients. The use of this test in asymptomatic patients during the Omicron wave had a statistically significant reduction in sensitivity compared with two reports of the same test in previous waves (13% vs. 86%, $p < 0.0001$; 13% vs. 83%, $p < 0.0001$). As SARS-CoV-2 continues to mutate, the sensitivity of RATs are altered and needs to be continually re-evaluated for each variant of concern if they are to be used as part of an infection control strategy.

**Keywords:** emergency; department; COVID; rapid; PCR; coronavirus; testing; asymptomatic

## 1. Introduction

The Coronavirus Disease 2019 (COVID-19) pandemic began in December 2019 and reached Australia on the 25 January 2020 [1]. This pandemic is the deadliest international public health emergency in history with a population-weighted infection fatality ratio between 1 and 2% in Australia (0.5% to 2.5% internationally) [2].

Variants of SARS-CoV-2 exist due to mutations in the protein target of the genome that alters the structure of the viral protein [3]. This virus is known to evolve at a rate of approximately $1.1 \times 10^{-3}$ substitutions per site per year, or one substitution every 11 days [4]. Although most mutations of SARS-CoV-2 have no perceivable impact, some raise concerns. The World Health Organization (WHO) developed a classification system to identify mutations that were either variants of interest (potential increase in disease severity and/or transmissibility) or variants of concern (strong evidence for increase in disease severity and/or transmissibility) [5]. For ease of communication, the WHO further recommended the use of the Greek Alphabet to identify novel variants [5].

The international approach to the COVID-19 pandemic has generally been to identify and contain severe acute respiratory syndrome coronavirus 2 (SARS-CoV-2) where possible. The early detection and quarantining of identified cases have been key to this strategy [6]. Reverse transcriptase polymerase chain reaction (RT-PCR) is considered the gold standard for the detection of SARS-CoV-2 [7,8]. This approach requires specialized laboratory facilities and requires time to process. A variety of rapid point-of-care antigen and molecular-based tests for the diagnosis of SARS-CoV-2 infection have been developed

and widely employed given their modest test characteristics [6]. The Abbott Panbio AG rapid antigen test (RAT) is one such available test. The manufacturer considers the use of the RAT in asymptomatic patients as 'off-label', that is, considered outside of the intended use of the product given the internal testing characteristics [9].

Given the evolving nature of the COVID-19 pandemic, many interim solutions were sought to minimize the spread. This included using tests outside of their intended use, including RATs in asymptomatic patients. Two studies have sought the validity of this particular RAT in a variety of asymptomatic populations and found sensitivities between 80.0% (61.4–92.3) to 86.67% (69.2–96.2) [10,11]. Both studies were performed prior to the Omicron wave of the COVID-19 pandemic [10,11].

The Panbio RAT is an immunochromatographic assay used to detect the SARS-CoV-2 nucleocapsid protein which requires no specialized instruments. Substitutions in this nucleocapsid protein and the addition of spike protein mutations in viral variants may therefore lead to a higher degree of false negative results compared with RT-PCR [12].

Considering the increasing use of RATs for screening and surveillance, we sought to evaluate the sensitivity of the Abbott Panbio RAT in asymptomatic patients during the Omicron wave of the COVID-19 pandemic (study period 15 January 2022 to 29 January 2022) to understand if the test characteristics were comparable between different variants of concern for SARS-CoV-2.

## 2. Materials and Methods

This study was a retrospective cohort study undertaken at an Australian tertiary emergency department (ED) that cared for 75,133 adult and pediatric presentations in 2021. The closest matching COVID-19-reported period (17 January 2022 to 23 January 2022) to the study period (15 January 2022 to 29 January 2022) identified 2189 confirmed cases of COVID-19 in the state in which this study was performed with a prevalence of 404.8 per 100,000 population [13]. For the same reporting period, the state had a total number of 1,157,078 COVID-19 vaccines administered, representing >99% of people aged 12 and over having had at least one dose and 96.1% of people aged 12 and over who had two or more doses [13]. For the reporting period, for all Australians, 1.0% (6267) of positive cases (by RT-PCR) were sequenced, with the majority (>99%) sequenced for B1.1.529 (Omicron) and the remainder B.1.617.2 (Delta) [13]. In Australia, the date for the start of the Omicron wave based on sequencing was 15 December 2021 [13].

All patients presenting to the ED over the two-week study period were included if they did not display COVID-19 symptoms (based on a standardized triage assessment form), were not considered close contacts, and received both tests. Patients were excluded if they had COVID-19 symptoms, had a positive RAT or RT-PCR in the last 28 days, if the RAT testing was not performed by a credentialled staff member, or if there were missing data.

Patients who were symptomatic were not tested using an RAT as infection control precautions were taken presuming that they were positive and therefore a negative RAT would not have affected the patient's infection precaution status.

Staff performing RATs on asymptomatic patients were predominantly nursing assistants, employed solely to perform RATs. However, other clinicians were also trained to perform RATs and included ambulance officers and registered nurses. All staff performing RATs in the department were required to undertake a standardized training and credentialling process for sampling in both tests. For both tests, a nasopharyngeal sample was taken as this is considered the most reliable source [14].

Data were collected from the medical record for all presentations to the ED within the study period. Data entry was undertaken by junior researchers and internally validated by a senior member of the research team with interrater reliability calculated using Cohen's kappa (κ).

Sensitivity and specificity with 95% CIs for the Panbio test were calculated using RT-PCR as the reference test with a binomial exact test used to estimate uncertainty and point estimates. Data were analyzed using STATA 16.0 [15].

To describe sensitivity patterns across different waves of the pandemic, we also compared the results of our study to the only two previous off-label studies of the Panbio RAT in asymptomatic patients [10,11]. A binomial exact test was used to calculate the probability of the point estimate in the comparison studies being drawn from the binomial distribution corresponding to sensitivity (x = true positives, *n* = all positives) in our study. The alternative hypothesis being that the sensitivity in our study is not equal to the point estimate in the comparator study.

The study was conducted in accordance with the Declaration of Helsinki and approved by the Ethics Committee of the University of Tasmania (project code 26899, 1 February 2022).

## 3. Results

Within the study period (15 January 2022 to 29 January 2022), 2059 patients presented to the ED and were screened for SARS-CoV-2. Patients were screened for exclusion criteria (*n* = 531) and removed. Of the 1528 remaining patients, 416 had either missing data or equivocal results (Figure 1). The final study population was 1114. The final study population (Table 1) included 569 males and 545 females with a mean age of 40.43 (±23.44), with 11.57% less than 14 years and 18.40% greater than or equal to 65 years. The majority of the RAT tests were performed by nursing assistants (784, 70.38%), with the remainder performed by either a registered nurse (93, 8.3%) or ambulance officer (186, 16.7%), with 51 (4.58%) not having their title documented. For all patients within this cohort, the median ED length of stay was 213 min (210 IQR) with 402 (36.09%) admitted.

Inter-rater reliability was tested using Cohen's kappa, calculated on a subset of observations (*n* = 51) for variables that were manually transcribed from digital medical record documents; screening, Panbio RAT, RT-PRC, and Tester returned kappa ranging from 0.857 to 1.0 (Table 2).

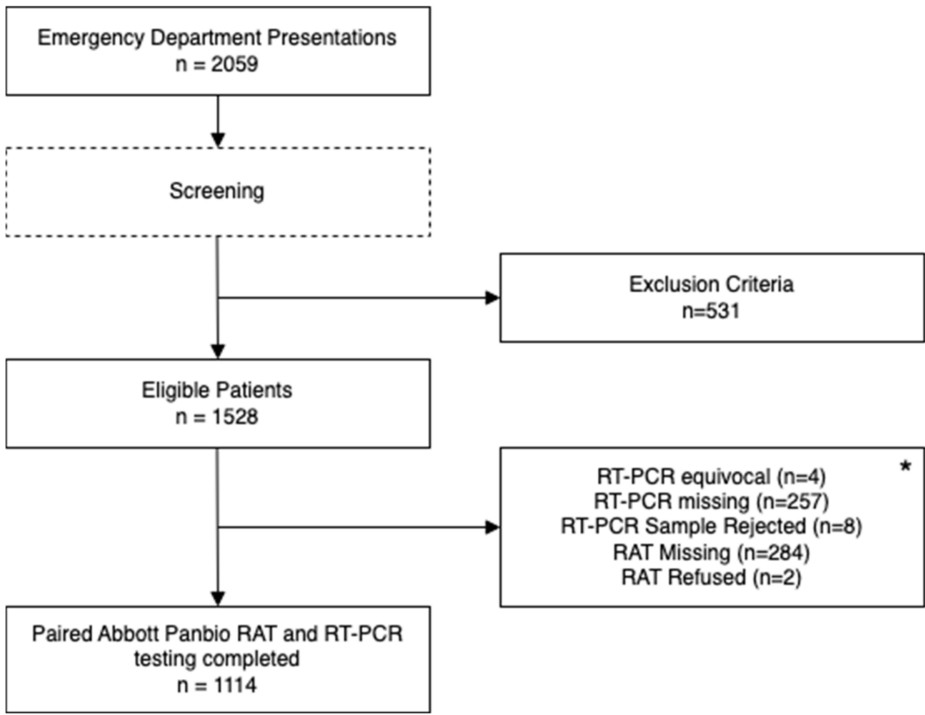

**Figure 1.** Screening process for presentations and paired RAT/RT-PCR samples. * Some excluded patients met more than one criterion for exclusion.

**Table 1.** Characteristics of patients who presented to the ED without COVID-19 symptoms and had an RAT and RT-PCR result.

| | RAT Result | | | | Total |
|---|---|---|---|---|---|
| | Positive | | Negative | | |
| | **True** | **False** | **True** | **False** | |
| Demographics | 3 | 2 | 1089 | 20 | 1114 |
| Gender, *n* (%) | | | | | |
| Male | 1 (33) | 1 (50) | 556 (51) | 11 (55) | 569 (51) |
| Female | 2 (66) | 1 (50) | 533 (48.9) | 9 (45) | 545 (49) |
| Age | | | | | |
| Mean (years, ±SD) | 40.66 (33.5) | 51.0 (7.1) | 40.5 (23.5) | 35.75 (21.2) | 40.43 (23.4) |
| Number of patients < 14 years old, *n* (%) | 1 (33.3) | - | 126 (11.6) | 2 (10) | 129 (11.6) |
| Number of patients ≥ 65 years old, *n* (%) | 1 (33.3) | - | 202 (18.5) | 2 (10) | 205 (18.4) |
| Emergency Department | | | | | |
| Australian Triage Category | | | | | |
| 1, *n* (%) | - | | 1 (0.1) | - | 1 (0.1) |
| 2, *n* (%) | 1 (33.3) | 1 (50) | 128 (11.8) | 2 (10) | 132 (11.9) |
| 3, *n* (%) | 1 (33.3) | 1 (50) | 447 (41.1) | 9 (45) | 457 (41.0) |
| 4, *n* (%) | 1 (33.3) | - | 405 (37.2) | 9 (45) | 415 (37.3) |
| 5, *n* (%) | - | - | 108 (9.9) | - | 109 (9.8) |
| Mode of Arrival | | | | | |
| Private Vehicle, *n* (%) | - | 1 (50) | 105 (9.6) | 3 (15) | 109 (9.8) |
| Walking, *n* (%) | 2 (66.7) | - | | 13 (65) | 643 (57.7) |
| Ambulance, *n* (%) | 1 (33.3) | 1 (50) | 307 (28.2) | 2 (10) | 311 (27.9) |
| Police, *n* (%) | - | - | 12 (1.1) | 1 (5) | 13 (1.2) |
| Public Transport, *n* (%) | - | - | 1 (0.1) | - | 1 (0.1) |
| Other, *n* (%) | - | - | 36 (3.3) | 1 (5) | 37 (3.3) |
| ED Length of Stay (min), median (IQR) | 323 (732) | 319 (377) | 211 (209) | 274 (282) | 213 (210) |
| Disposition | | | | | |
| Discharged, *n* (%) | 2 (66.7) | 2 (100) | 694 (63.7) | 14 (70) | 712 (63.9) |
| Admitted, *n* (%) | 1 (33.3) | - | 395 (36.3) | 6 (30) | 402 (36.1) |
| RAT Tester | | | | | |
| Registered Nurse, *n* (%) | - | - | 89 (8.2) | 4 (20) | 93 (8.4) |
| Assistant in Nursing, *n* (%) | 2 (66.7) | - | 767 (70.4) | 15 (75) | 784 (70.4) |
| Ambulance Officer, *n* (%) | - | 2 (100) | 184 (16.9) | | 186 (16.7) |
| Other, *n* (%) | 1 (33.3) | - | 49 (4.5) | 1 (5) | 51 (4.6) |

**Table 2.** IRR for variables: screening, Panbio RAT, RT-PRC, and Tester calculated for subset *n* = 51 observations.

| Variable | Agreement | κ | *p* |
|---|---|---|---|
| Screening | 100% | 1.0 | <0.0001 |
| PanBio RAT | 96.08% | 0.8571 | <0.0001 |
| RT-PCR | 96.08% | 0.8817 | <0.0001 |
| Tester | 94.12% | 0.8947 | <0.0001 |

The Panbio RAT sensitivity was 13% (95% CI 0.028, 0.336) with a positive predictive value of 0.6 (95% CI 0.147, 0.947) and negative predictive value of 0.982 (95% CI 0.972, 0989) (Table 3).

When comparing the sensitivity of the Panbio RAT during the Omicron wave against the asymptomatic patient populations in previous waves, there was a statistically significant difference between sensitivity: Tran vs. Winkel (13% vs. 86%, *p* < 0.0001), Tran vs. Aranaz-Andres (13% vs. 83%, *p* < 0.0001) (Table 4).

**Table 3.** Paired results of RAT using RT-PCR, sensitivity, specificity, and NPV PPV for RAT in asymptomatic patients presenting to emergency department.

|  | RT-PCR Positive | RT-PCR Negative |
|---|---|---|
| Panbio positive | 3 | 2 |
| Panbio negative | 20 | 1089 |
|  | Binomial exact [95% conf] | |
| Incidence (%) | 2.1% [1.3, 3.08] | |
| Sensitivity | 0.13 [0.027, 0.336] | |
| Specificity | 0.99 [0.993, 1.0] | |
| Positive predictive value | 0.6 [0.147, 0.947] | |
| Negative predictive value | 0.982 [0.972, 0.989] | |

**Table 4.** Binomial exact test for the comparison between current study Tran et al. and Winkel et al. 2021 and Aranaz-Andres et al. 2021, all of which evaluated the performance of Abbott Panbio RAT in asymptomatic patients in periods prior to the Omicron wave.

| Study of Interest (x, n) | Comparative Study (est) | *p*-Value |
|---|---|---|
| Tran (3, 23) | Winkel (0.86) | <0.0001 |
| Tran (3, 23) | Aranaz-Andres (0.83) | <0.0001 |
| Aranaz-Andres (25, 30) | Winkel (0.86) | 0.6015 |

## 4. Discussion

Public health responses attempting to control the COVID-19 pandemic take a multi-faceted approach. Internationally, this has included movement restrictions, disease modelling, novel vaccine development, and adequate infection prevention and control frameworks, of which RATs are part of [16].

The early identification and isolation of COVID-19 patients remains central to the international pandemic response to provide adequate infection prevention and control to limit spread [17]. The purpose of RATs as part of a screening tool is to identify and isolate those with the disease from those without, thereby limiting the spread [18]. This has particular significance for the COVID-19 pandemic given the highly transmissible nature of SARS-CoV-2 [19]. The R number for COVID-19 has been driving policy decisions since the start of the pandemic [20]. $R_0$ represents the basic reproduction number, that is, the number of people each infected person will infect on average if there is no pre-existing immunity in the community [20]. $R_e$ represents the effective reproduction number, or the number of people that can be infected by one person at any specific time and therefore changes based on population immunity [21]. Pathogens with a high $R_0$ score have the potential to cause larger epidemics or pandemics [20]. The $R_0$ for SARS-CoV-2 was estimated to be around 3, and, as has been found, is highly transmissible and severe enough to outstrip first-world health resources and require an aggressive and effective public health response [22]. Given the highly transmissible nature of SARS-CoV-2, the failure to detect it in asymptomatic patients consequently means that they are cohorted with patients who are not infected, thereby enabling a growth advantage and significantly increasing the spread and overwhelming resources [23].

Although RT-PCR remains the gold standard, it requires a substantial amount of finances and resources. The process time required from test to result is also long. In contrast, the low-cost, readily available, and fast results of RAT tests have been critical to the international pandemic response [6]. The COVID-19 pandemic has seen mutations to SARS-CoV-2 with each wave, with the Omicron wave showing the greatest change with 72 mutations (with at least 30 on the nucleocapsid protein), and more patients presenting without symptoms when compared with its predecessor waves [10,11].

It has been postulated that a reduction in sensitivity exists when a patient is asymptomatic due to differences in viral load distribution [24–26]. For this reason, and as a result

of internal manufacturer testing, RAT testing remains off-label for asymptomatic patients. Studies investigating the off-label use of RATs during previous waves of the pandemic have shown moderate (>80%) sensitivity [10,11].

The World Health Organization defines screening as the presumptive identification of unrecognized disease or defects by means of tests, examinations, or other procedures that can be applied rapidly [27,28]. The screening is intended for all persons in an identified population who do not display symptoms or have conditions from the disease for which they are being screened for [29]. There are both resource and ethical obligations when developing screening tools to maximize benefit and minimize harm. For COVID-19, screening can reduce the spread and resultant resource requirements to minimize morbidity and mortality. Harm associated with screening in general is also relevant to COVID-19. For COVID-19, the harm of a false negative can result in outbreaks and overwhelmed resources, whereas false positives have implications on unnecessary isolation, including work and travel.

Wilson and Junger developed the WHO principles of screening which mandates tests to be highly specific, highly sensitive, validated, safe with a high positive and negative predictive value, and acceptable to the target population [30]. The use of RATs as a screening tool requires constant consideration to determine if it meets the WHO principles of a screening test. As disease prevalence increases, the sensitivity of the RAT becomes increasingly important given the absolute numbers of positive cases that are missed and therefore the higher risk of overwhelming health resources through outbreaks. Far reaching implications may exist if RAT results are inappropriately used or interpreted in vulnerable areas such as hospitals and aged care facilities.

Although previous studies have explored the sensitivity of RATs in asymptomatic patients, our study is the first known publication to report on the sensitivity of the Panbio RAT during the initial Omicron wave. Our results have shown a sensitivity of only 13% (95% CI 0.028, 0.336), falling well below the WHO requirement for a diagnostic test (sensitivity of 80% or greater) [31].

Previous studies have not specifically compared sensitivity across different COVID-19 waves to understand the reproducibility of test characteristics for off-label use. Our study has found significant digression from the test characteristics of the Panbio RAT when compared with other waves of the COVID-19 pandemic. Whereas sensitivity in previous waves was reported at 86% and 83%, we found that it was only 13% in our population during the Omicron wave ($p < 0.001$) [10,11]. As previously stated, one hypothesis for this is the significant degree of mutations compared to other WHO variants of concern. Another factor that may contribute to this difference is due to the high level of vaccinations in the population tested (>99% of people aged 12 and over had at least one dose and 96.1% of people aged 12 and over had two or more doses) [13]. The premise of vaccination has focused on limiting the spread through a reduction in transmission [32]. During the delta wave, reports showed that highly vaccinated populations did not reduce the sensitivity of the RAT, although no reports during the initial Omicron wave have been reported [33].

Comparing the same RAT for different mutations of SARS-CoV-2 highlights the need to be vigilant regarding the off-label use of any test. With the various waves of COVID-19 underpinned by ongoing unpredictable mutations to SARS-CoV-2, this effect on test characteristics must be taken into consideration if the Panbio RAT is to be relied upon as a screening tool. Furthermore, test characteristics need to be re-evaluated with each variant of concern.

It is also known that sensitivity of RATs and RT-PCR varies during an illness, with sensitivity starting low and peaking at day 4 of illness [34]. Furthermore, by day 6 of a COVID-19 illness, RT-PCR continues to remain sensitive (86%), yet RAT sensitivity falls to 61% [34]. Given that the study population in this instance was asymptomatic, it is difficult to gauge what day of infection patients were currently on and is therefore less important in this context.

The Australian Government has invested AUD 250 million for the continued distribution of RATs to residential aged care facilities, aboriginal controlled community health services, general practitioner-led respiratory clinics, and remote and rural communities where there is limited access to testing [35]. Furthermore, the government has funded AUD 267 million to deliver RATs to supported independent living residential disability care services [35]. With such significant investment in RATs, it is important that they are used economically, and, to do so, they need to be performed within the tests' ability to be inclusive of sensitivity for the various SARS-CoV-2 variants.

Despite the identified broad variability in the sensitivity of RATs in asymptomatic patients as part of a screening tool in the public health response to recognize and contain COVID-19 individuals, Australia avoided the worse effects of the COVID-19 pandemic [16]. This is likely to have been supplemented by other strategies to contain the virus, including strict lockdown, border closures, and quarantine requirements [36]. Part of this containment included isolating asymptomatic close contacts, thereby negating the need for an RAT, as these patients were treated as if they were infected with SARS-CoV-2 from an infection control viewpoint.

## 5. Limitations

One limitation in this interpretation involved the cohorts studied. The population studied by Winkel et al. were predominantly males who were often young professional athletes [10]. Aranaz-Andres et al. studied a population of hospitalized patients who, although were asymptomatic, were screened as part of a hospital outbreak protocol defined as the presence of three or more epidemiologically linked cases confirmed with RT-PCR in that area [11]. Our study included a much wider population of asymptomatic patients, but also had limitations in that they were presenting to the emergency department for an acute health concern.

Another limitation was the ability to compare the sensitivity of the Panbio RAT in symptomatic patients with and without COVID-19 upon RT-PCR. The Panbio RAT was used to rationalize the cohorting of patients with suspected COVID-19 to minimize transmission. The hospital considered all patients who were symptomatically treated as if they had COVID-19 unless proven otherwise. As such, a negative RAT would not have changed how symptomatic patients were managed and was therefore not performed. Given that the scope of our study was to assess asymptomatic patients only, the lack of a symptomatic comparison cohort does not affect the conclusion that there is variability with RAT sensitivity in asymptomatic patients.

Our evaluation was limited to a single brand of the rapid antigen test and therefore validity and applicability using other brands has not been tested. Given that different brands have used the same immunochromatographic assay to detect the SARS-CoV-2 nucleocapsid protein, it is postulated (but not proven) that these results may be applicable to other brands.

The cycle threshold (CT) for PCR tests was not collected prior to the de-identification of the data. This information can be important to assist in determining causation for the high level of false negative RAT results for asymptomatic patients. However, given that CTs are influenced by a number of factors, including the PCR test kit, when the sample was collected, the machines used for testing, and the clinician testing technique and sampling methods, the results would be difficult to replicate. Moreover, the CT results, although perhaps indicating a cause for the results, does not change the false negative rate of RATs in asymptomatic patients in our cohort.

## 6. Conclusions

A consideration of test effectiveness must be given to the off-label use of RATs, particularly when used as a screening tool. We have shown that, with changes in viral structure, the sensitivity of the Panbio test is affected. For the Omicron variant of SARS-CoV-2, the sensitivity of asymptotic patients decreased from >80% to 13%. As SARS-CoV-2 continues

to mutate, the effectiveness of RATs needs to be evaluated for each variant of concern if they are to be used as part of an infection control strategy.

**Author Contributions:** Conceptualization, V.T., B.D. and L.C.; methodology, V.T. and B.D.; data collection, G.B., A.L. and S.W.; formal analysis, G.B., V.T. and Z.A.; writing—original draft preparation, V.T.; writing—review and editing, V.T., G.B., B.D., A.L., S.W. and L.C. All authors have read and agreed to the published version of the manuscript.

**Funding:** This research received no external funding.

**Institutional Review Board Statement:** The study was conducted in accordance with the Declaration of Helsinki and approved by the Ethics Committee of the University of Tasmania (project code 26899, 1 February 2022).

**Informed Consent Statement:** Patient consent was waived due to the low-risk nature of the study.

**Data Availability Statement:** The data presented in this study are available on request from the corresponding author. The data are not publicly available due to privacy laws in the country of origin.

**Conflicts of Interest:** The authors declare no conflict of interest.

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
