# Peer review of "Evaluation of the Abbott Panbio™ COVID-19 Ag Rapid Antigen Test for Asymptomatic Patients during the Omicron Wave"

_2571-8800, doi:10.3390/j6010015_

Round 1

Reviewer 1 Report

The manuscript by Tran et al. evaluated the Abbott PanbioTM COVID-19 Ag rapid antigen test (RAT) for asymptomatic patients. This is a very synthetic and weak paper. Considering that data were analyzed over a year ago, they are not representative of the current pandemic situation. For these reasons, the paper is not suitable for publication in its present form.

Major concerns:

-       There is no description of sampling. No information about the number and type of samples or the time interval between sampling and tests. This part should be mandatory in this type of paper.

-       It is not clear whether the study was reviewed and approved by an independent ethical review committee. The authors should clearly mention the ethical approval in the manuscript.

-       The population of the study is not characterized. There is no information about age and sex distribution (possible bias).

-       The data are not representative of the current pandemic situation.

-       The data analysis was minimal. Although the targets of the study were asymptomatic patients, it would have been necessary to include data from samples of symptomatic cases for the timely comparison of sensitivity, specificity, PPV, NPV, etc. These data do not adequately support the authors' conclusions.

-       Tables 2 and 3 should be revised into a single table.

-     Omega prevalence was also not mentioned. Considering the aim of the paper, it would have been appropriate to characterize the variant of positive patients.

-       The author must include the range of Ct values obtained from real-time PCR to better understand and discuss data from RAT.

-       The discussion is very weak.

Reviewer 2 Report

In this article authors discuss about the relevance of RAT in identifying COVID 19 patients. I have the following recommendations

1. In methodology include, prospective/retrospective study?

2. Was IRB clearance obtained

3. How was the test obtained, site, sample, nurse/physician, adequate sample?

4. Did the authors have the sensitivity of this test in symptomatic patients?

5. Add few more relevant citations. example PMID: 32267941

Reviewer 3 Report

In Table 2 there are actually 20 cases that are positive as per the gold standard PCR assay but negative on the Panbio RAT assay, yet the description above table2 states the opposite. Please correct this statement to reflect these 20 as false negatives because the gold standard assay found these to be positive.

Original sentence: The Panbio RAT detected 20 cases of SARS-CoV-2 in 93 asymptomatic patients where the RT-PCR was positive (Table 2).

The authors state that there are 72 new mutations in the omicron strain, please clarify how many of them are in the Nucleocapsid protein which is what is being measured in the Panbio RAT assay.

The study is under-referenced and could be better if a strong conclusion was presented rather than merely stating your observations from this cohort. A similar article was published, but was not referenced: title: Evaluation of the Abbott Panbio COVID-19 Ag rapid antigen test for the detection in asymptomatic Canadians. There are several other meta analyses presented such as in the reference article titled: Performance of Rapid Antigen Tests for COVID-19 Diagnosis: A Systematic Review and Meta-Analysis.

Author Response

In Table 2 there are actually 20 cases that are positive as per the gold standard PCR assay but negative on the Panbio RAT assay, yet the description above table2 states the opposite. Please correct this statement to reflect these 20 as false negatives because the gold standard assay found these to be positive.

Original sentence: The Panbio RAT detected 20 cases of SARS-CoV-2 in 93 asymptomatic patients where the RT-PCR was positive (Table 2).

Thank you for highlighting this - we have corrected the statement

The authors state that there are 72 new mutations in the omicron strain, please clarify how many of them are in the Nucleocapsid protein which is what is being measured in the Panbio RAT assay.

Thank you for highlighting this - we have added in "with at least 32 in the nucleocapsid protein"

The study is under-referenced and could be better if a strong conclusion was presented rather than merely stating your observations from this cohort. A similar article was published, but was not referenced: title: Evaluation of the Abbott Panbio COVID-19 Ag rapid antigen test for the detection in asymptomatic Canadians. There are several other meta analyses presented such as in the reference article titled: Performance of Rapid Antigen Tests for COVID-19 Diagnosis: A Systematic Review and Meta-Analysis.

Thank you for highlighting this - we have undergone major revisions and now have 32 references. We note similiar papers with similiar scope. Our intention was to highlight the variability in sensitivity across variants of concern, which others have only touched on. We feel that this is significant because in many juristictions, there is still a policy to RAT test if asymptomatic but a close contact in order to work or enter a high risk area. 

Round 2

Reviewer 1 Report

The validation of a rapid test is precisely to understand its usefulness during a pandemic or phase thereof. The structure of the manuscript does not reflect the objectives stated in the responses. I do not believe that the evaluation of only one rapid test in a very limited period during the pandemic can allow one to say that "Our goal was....(to) identify the utility at the time so that for future pandemics." This statement seems very forced. After all, the general limitations of a rapid test are well known.

Moreover, the absence of CT values does not allow us to understand the cause of such poor results of this rapid test.

Consequently, the conclusions are not supported by the data shown.
